# A Combined Homing Trajectory Optimization Method of the Parafoil System Considering Intricate Constraints

**Weichao He [1], Jiayan Wen [1], Jin Tao [2,3,*] and Qinglin Sun [2]**

[1] School of Electrical Electronics and Computer Science, Guangxi University of Science and Technology, Liuzhou 545006, China; weichaoh1993@163.com (W.H.); wenjiayan2012@126.com (J.W.)
[2] College of Artificial Intelligence, Nankai University, Tianjin 300350, China; sunql@nankai.edu.cn
[3] Silo AI, 00100 Helsinki, Finland
[*] Correspondence: taoj@nankai.edu.cn

**Abstract:** In order to achieve an accurate airdrop in the actual environment, the influence of complex interferences, such as wind field and the terrain of the environment, must be taken into account. Aiming at this problem, a combined trajectory planning strategy of a parafoil system subjected to intricate conditions is proposed in this paper. This method divides the parafoil airdrop area into an obstacle area and a landing area, then, considering the terrain environment surface, a model for the parafoil system in the wind field is built in the obstacle area. The Gauss pseudo-spectral method is used to transform the complex terrain environment constraint into a series of nonlinear optimal control problems with complex constraints. Finally, the trajectory of the landing area is designed by means of multiphase homing, and the target parameters are solved by the improved marine predator algorithm. The simulation results show that the proposed method has better realizability than a single homing strategy, and the optimization results of the improved marine predator algorithm have higher accuracy.

**Keywords:** parafoil system; trajectory planning; intricate constraints; Gauss pseudo-spectral method; multiphase homing; marine predator algorithm

## 1. Introduction

The parafoil system is a kind of flexible aircraft with good gliding, controllability, and stability. Unlike the traditional parachute system, which can only move with the wind and is not controlled [1], the parafoil system can maintain the airfoil and provide lift and forward force by relying on the air chamber formed by ram air after parachute opening [2]. It can also change the attitude of the parafoil in the air by pulling down the control rope, thereby precisely controlling the heading of the whole system. With the continuous enhancement of China's comprehensive national strength, parafoils have played an essential role in aerospace, disaster relief, fixed-point airdrops, and other fields with their unique structure and significant advantages of in-flight stability [3] carrying capacity, and excellent aerodynamic characteristic's irreplaceable role [4].

The homing strategies of the parafoil system in the actual airdrop environment are roughly divided into two types: the optimal-control homing strategy considering the constraints of a complex environment and the multiphase homing strategy without considering the limitations on obstacles. Chen et al. [5] proposed an optimal segmented-path planning method for parafoil based on the gradient descent method, which transforms the multi-objective optimization problems, e.g., accurate landing against the wind, low-control-energy consumption, and obstacle avoidance into weighted single-objective-optimization problems. Luo et al. [6] and Figueroa et al. [7] transformed the optimal control trajectory planning into a nonlinear programming problem by using the pseudo-spectral method and then solved it by using the sequential quadratic programming algorithm. Aiming at the problem that the target trajectory planned by the traditional particle model is challenging to

meet the system dynamics constraints in a complex environment, SUN [8] introduced the six-degrees-of-freedom dynamics model of the parafoil system into the trajectory planning of parafoil homing and designed a third-order trajectory optimization strategy based on segment-point planning, discrete-point initial planning, and discrete-point self reconstruction by improving the Gauss pseudo-spectrum method. Cardoso [9] divided the homing trajectory of the parafoil system into three segments and optimized it with an improved genetic algorithm; Tao [10,11] used a chaotic particle swarm optimization algorithm and quantum genetic algorithm to optimize the homing trajectory of the parafoil system; Messai [12] took the minimum energy consumption as the objective function, time-consuming and turning radius as the input variables, established the optimization model of five-segment homing trajectory, and optimized the parameters of each segment by using the Gauss pseudo-spectral method.

The research of the above scholars focused on the homing strategy of the parafoil system. In the real airdrop environment, there is no obstacles in a small range around the target point. The combined homing approach enables the parafoil system to land on the target point more accurately. Based on this idea, Cho [13] roughly divided the parafoil airdrop environment into the obstacle and landing areas. When the parafoil system is in the obstacle area, the fast-search random-tree algorithm is used to perform path search and obstacle avoidance and then enter the landing area. After that, the trajectory is designed through the multiphase homing strategy, and the target parameters are solved using the genetic algorithm. Its research results can meet the needs of parafoil airdrop in complex environments. Nonetheless, there are also some limitations, such as the fact the influence of the wind field is not considered during the homing process of the parafoil, and the traditional genetic algorithm used in the landing area is easy to fall into local optimization.

In this paper, the idea of dividing the complex airdrop area into obstacle and landing areas was adopted. The wind-field disturbance and the feasible area after obstacle avoidance were considered in the obstacle area. In the obstacle area, the Gauss pseudo-spectral method was used to optimize the parafoil homing trajectory with the control quantity as the optimization goal. In the landing area, the multiphase homing method was used, and the marine predator algorithm with higher convergence was applied to optimize the trajectory. The simulation results showed that the proposed method has good adaptability and effectiveness in homing trajectory planning for parafoil systems.

The rest of this paper is organized as follows: Section 2 formulates the homing trajectory planning problem; Section 3 describes the Gauss pseudo-spectral method and the marine predator algorithm used for optimization; Section 4 provides the simulation results to illustrate the effectiveness of the method; and Section 5 concludes this paper.

## 2. Problem Formulation

### 2.1. Parafoil System Particle Model

The parafoil system is usually composed of an umbrella body and load. If the umbrella body and the load are regarded as a rigidly connected whole, a 6-DOF (degrees of freedom) rigid-body model can be established. By further analyzing the relative motion between the parafoil and the payload, the dynamic models of the parafoil system with 7-DOF, 8-DOF, and higher degrees of freedom can be established. However, the higher the degree of freedom of the model, the higher the computational complexity [14]. Therefore, when studying the homing trajectory of the whole parafoil system, it is not necessary to consider the relationship between the parafoil body and the load. The entire system can be regarded as a particle, and the parafoil particle model can be used to replace the complex high-degree-of-freedom model to realize trajectory planning.

For the parafoil system, the change of bilateral downward deflection has little effect on its vertical velocity and glide ratio. The change of unilateral downward deflection impacts its vertical velocity, glide ratio, and roll angle. Therefore, the control of unilateral downward deflection is usually limited to the range of small and medium deflection (0–50%) [15]. As a result, the parafoil system's stable-flight speed and glide ratio can be regarded as a constant

under any control and within a certain lower deflection range [16]. In order to simplify the model, assumptions can be made as follows [17–19]:

- When the parafoil system is in the stable-descent stage, it can be regarded as a particle, and the vertical and horizontal velocities of the system remain unchanged;
- There is no delay in the influence of the parafoil in terms of control inputs;
- The wind field is known, and the influence of wind speed and direction can be transformed into the position offset of the initial point.

In this paper, the plane geodetic system was selected as the coordinate system, the coordinate origin was the target point, and the motion equation of the parafoil system can be expressed as:

$$\begin{cases} \dot{x} = v_{\text{s}} \cos \varphi + v_{\text{w},x} \\ \dot{y} = v_{\text{s}} \sin \varphi + v_{\text{w},y} \\ \dot{\varphi} = u \\ \dot{z} = v_z \end{cases}, \tag{1}$$

where $(x, y)$ is the horizontal position of the parafoil system, $z$ is the vertical height, the unit of position and height is m; $v_{\text{s}}$ is the horizontal velocity of the parafoil system, $v_z$ is the vertical velocity of the parafoil system, $v_{\text{w},x}$ and $v_{\text{w},y}$ are the components of wind speed on the $x$ and $y$ axes, respectively, the unit of speed is m/s; $\varphi$ is the turning angle, the unit of angle is rad; $\dot{\varphi}$ is the turning angular acceleration, the unit is rad/s.

*2.2. Wind-Field Model*

The wind field mainly includes constant wind, gust, and flocculation wind [20]. In this paper, when the parafoil system was in the obstacle area, due to the harsh-terrain environment and constraints, such as mountains, it was easy to be disturbed by the wind field with sudden and severe speed change. Therefore, the gust effect was considered as the main influence of the wind field on the parafoil system, and the NASA classic gust model was selected [21], as shown in Formula (2).

$$\begin{cases} v_{\text{wind}} = \frac{v_{\text{max}}}{2} \left( 1 - \cos \frac{t}{100} \pi \right) & 0 \leq t \leq 100 \text{ s} \\ v_{\text{wind}} = v_{\text{max}} & 100 \text{ s} < t \leq 200 \text{ s} \\ v_{\text{wind}} = \frac{v_{\text{max}}}{2} \left( 1 - \cos \frac{t-300}{100} \pi \right) & 200 \text{ s} \leq t \leq 300 \text{ s} \end{cases}, \tag{2}$$

where $v_{\text{wind}}$ is the wind speed, and the maximum wind speed is set as $v_{\text{max}} = 4$ m/s, the direction is the positive direction of the $x$ axis. $t$ is time. When $0 \leq t \leq 100$ s and $200 \text{ s} \leq t \leq 300$ s, the parafoil system is affected by two kinds of gusts. However, when $100 \text{ s} < t \leq 200$ s, the parafoil system is affected by constant wind.

When the parafoil system enters the landing area, the terrain is relatively flat, and the constantly insignificant wind affects the flight trajectory of the parafoil system. At this time, the constant-wind impact is selected as the main impact of the wind field on the parafoil.

*2.3. Complex-Terrain Model*

When the parafoil system passes through the obstacle area, it encounters terrain environments, such as peak areas and no-fly areas. At this time, it is necessary to avoid these threatening environments. In this paper, the mountain was designed as a series of concentric circles with $(x_p, y_p)$ as the center, and the terrain environment can be expressed as:

$$\| (x - x_p, y - y_p) \|_2 < R_p, \quad p = 1, 2, \cdots, \tag{3}$$

$$z = h_0 \sin \left( \frac{\pi}{\delta} \left( R_p - \sqrt{(x - x_p)^2 + (y - y_p)^2} \right) \right), \tag{4}$$

where $R_p$ is the maximum mountain radius, $h_0$ denotes the height of the mountain, the unit of the mountain radius and height of the mountain is m, $\delta$ is the smooth coefficient of the mountain, and $p$ denotes the serial number of the mountain.

### 2.4. Trajectory Planning Problems in a Complex Environment

In the actual airdrop environment, the parafoil system often encounters obstacles, such as high mountains and tall obstacles, but there is an open area in a certain range in the final landing stage [22]. Therefore, the homing environment of the parafoil can be artificially divided into two areas: the obstacle area and the landing area. Figure 1 is the schematic diagram of the homing trajectory planning of a parafoil system in a complex environment.

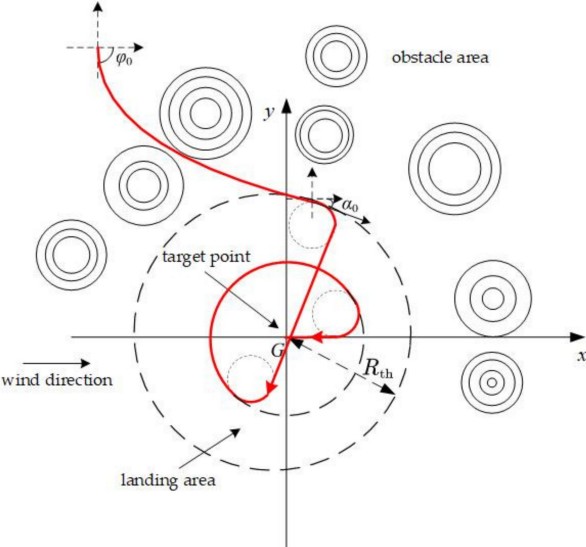

**Figure 1.** Schematic of a combined parafoil system for homing trajectory planning in a complex environment.

In Figure 1, the origin of the geodetic coordinate system is the G airdrop target landing point, and the circular area with the target point is the center and $R_{th}$ as the radius is the landing area without obstacles. There are many high-mountain-obstacle areas between the launch point and the landing area. Therefore, reasonable trajectory planning must be adopted so that the parafoil system can smoothly pass through the obstacle area and enter the landing area under the influence of the gust, and finally land accurately on the target point.

### 2.5. Obstacle Constraints

#### 2.5.1. Initial Value Constraint

If the initial time $t_0$ is known, the initial state can be expressed as:

$$\left\{ \begin{array}{l} x(t_0) = x_0 \\ y(t_0) = y_0 \\ z(t_0) = z_0 \\ \varphi(t_0) = \varphi_0 \end{array} \right. . \tag{5}$$

#### 2.5.2. Terminal Constraint

The target point of the obstacle area is the starting point of the landing area, and the terminal time $t_e$ is fixed. The position of the parafoil system after obstacle avoidance is regarded as the terminal point of the obstacle area, and its position deflection is treated as the terminal constraint of the obstacle area. In this way, the parafoil system can land on the target point with high precision.

$$\left\{ \begin{array}{l} t_e = z_0/v_z \\ x(t_e) = x_e \\ y(t_e) = y_e \\ z(t_e) = z_e \end{array} \right. . \tag{6}$$

### 2.5.3. Control Constraints

The maximum control quantity $u_{max}$ corresponds to the minimum turning radius. In this paper, the horizontal velocity is $v_s = 13.8$ m/s and the minimum radius is 100 m, then the maximum control value is $u_{max} = 0.138$.

$$|u| \leq u_{max}. \tag{7}$$

### 2.5.4. Real-Time Path Constraint

According to Formulas (3) and (4), the real-time constraint of terrain avoidance trajectory meets the following formula:

$$\| \left( x(t) - x_p, y(t) - y_p \right) \|_2 \geq R_p - \frac{\delta}{\pi} \arcsin \frac{z(t)}{h_p}. \tag{8}$$

### 2.6. Optimization Objective in the Obstacle Area

In the obstacle area, the parafoil system needs to avoid obstacles under the action of the wind field, so it is mainly considered to minimize the consumption of control quantity, which is expressed as:

$$J = \int_{t_0}^{t_e} u^2 dt. \tag{9}$$

Therefore, the trajectory optimization problem of the parafoil system in the obstacle area can be transformed into a kind of nonlinear optimal control problem with complex constraints [23] and then solved by the Gauss pseudo-spectrum method.

### 2.7. Trajectory Planning in the Landing Area

After the parafoil system successfully passes through the obstacle area, it enters the landing area. At this time, the initial state of the landing area is the end state of the obstacle area. Considering that there are almost no obstacles in the landing area, the parafoil system trajectory can be designed using the multiphase homing strategy, as shown in Figure 2, so as to improve the accuracy of its final landing and reduce the amount of control consumption.

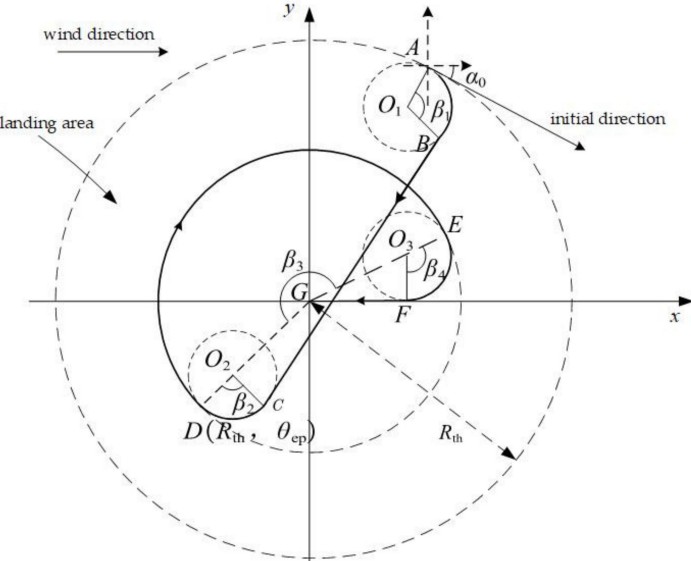

**Figure 2.** Schematic diagram of trajectory planning in landing area.

As shown in Figure 2, point A is the initial position of the parafoil system in the landing area and is also the end position of the parafoil system after the parafoil system completes obstacle avoidance in the obstacle area. Its state is shown in Formula (4). The multiphase homing trajectory planning in this paper is similar to the traditional three-stage homing

trajectory. The whole multiphase homing trajectory is roughly divided into three sections: the target-approach segment BC for the parafoil system to perform the gliding motion, the energy-control segment DE for the hovering and cutting heights, and the landing segment FG for the headwind alignment. Among them, CD and EF are transition sections, and the corresponding radius of the arc trajectory is the minimum turning radius $R_{\min}$ of the parafoil system, and the center angle corresponding to the arc meets the constraint of $0 < \beta_1, \beta_2, \beta_4 < \pi$.

The key to the optimization of the whole segment homing part is to determine the parameters $R_{\mathrm{ep}}$ and $\theta_{\mathrm{ep}}$, where $R_{\mathrm{ep}}$ is the radius corresponding to the circular trajectory of the energy control section and $\theta_{\mathrm{ep}}$ is the angle included between the connecting line between the entry point D and the target point G and the $x$ axis. In order to ensure the flight stability of the parafoil system in the energy-control section and considering the characteristics of the airdrop environment, the turning radius $R_{\mathrm{ep}}$ cannot be greater than the radius $R_{\mathrm{th}}$ of the landing area, so the range is set as $(R_1, R_2)$, where $R_1$ and $R_2$ are the upper and lower limits of the circular trajectory radius of the energy control section, and $R_2 = R_{\mathrm{th}}$, and the value range of $\theta_{\mathrm{ep}}$ is $(-\pi, \pi)$.

In order to ensure that the parafoil system achieves the smallest distance between the landing point and the target point in the landing area, and to meet the requirements of flared landing and avoid a collision with obstacles, the segmented trajectory objective function is set as:

$$J = \left| R_{\min}(\beta_1 + \beta_2 + \beta_4) + R_{\mathrm{ep}}\beta_4 + \| BC \| + \sqrt{(R_{\mathrm{ep}} - 2R_{\min})R_{\mathrm{ep}}} - fz_0 \right|, \quad (10)$$

where $R_{\min}(\beta_1 + \beta_2 + \beta_4)$ is total length of transition arcs AB, CD, and EF. $R_{\mathrm{ep}}\beta_3$ denotes the length of energy-control section DE, $\| BC \|$ is the length of the BC section, $\sqrt{(R_{\mathrm{ep}} - 2R_{\min})R_{\mathrm{ep}}}$ is the length of the FG section, $f$ is the glide ratio without the power plant, and $fz_0$ is the horizontal flight distance corresponding to the altitude when the parafoil system enters the landing area.

At the same time, in order to ensure that the range of motion of the parafoil system is limited to the landing area and is limited by the minimum turning radius, the radius of the landing area must also meet the following requirements: $R_{\mathrm{th}} \geq 2R_{\min}$.

In this paper, the coordinate of the parafoil system at point A is set as $(x_a, y_a, z_a)$, $\alpha_0$ is its initial flight direction angle, then point A on the horizontal plane can be expressed as:

$$\begin{bmatrix} x_a \\ y_a \end{bmatrix} = \begin{bmatrix} x_{\mathrm{e}} \\ y_{\mathrm{e}} \end{bmatrix}. \quad (11)$$

$O_1$ point coordinate is written as:

$$\begin{bmatrix} x_{O_1} \\ y_{O_1} \end{bmatrix} = \begin{bmatrix} x_a \\ y_a \end{bmatrix} + R_{\min} \begin{bmatrix} \cos \alpha_1 \\ \sin \alpha_1 \end{bmatrix}. \quad (12)$$

$O_2$ point coordinate is expressed as:

$$\begin{bmatrix} x_{O_2} \\ y_{O_2} \end{bmatrix} = (R_{\mathrm{ep}} - R_{\min}) \begin{bmatrix} \cos \theta_{\mathrm{ep}} \\ \sin \theta_{\mathrm{ep}} \end{bmatrix}. \quad (13)$$

$\angle BCx$ is the angle between BC and the $x$ axis, which can be expressed as:

$$\angle BCx = \begin{cases} \mathrm{sign}(y_{BC}) \cdot \frac{\pi}{2}, & x_{BC} = 0 \\ \frac{1 - \mathrm{sign}}{2} \cdot \mathrm{sign}(y_{BC}) \cdot \pi + \arctan \frac{y_{BC}}{x_{BC}}, & x_{BC} \neq 0 \end{cases}. \quad (14)$$

The length of the BC section is arranged as:

$$\| BC \| = \| O_1 O_2 \| = \sqrt{x_{O_1 O_2}^2 + y_{O_1 O_2}^2}. \tag{15}$$

Define the symbolic variable $s$ and set $s = -1$ when turning clockwise and $s = 1$ when turning counterclockwise. Therefore, the transition angles $\beta_1, \beta_2, \beta_3$ and $\beta_4$ can be expressed as:

$$\begin{cases} \beta_1 = s \cdot (\angle BCx - \alpha_0) \\ \beta_2 = s \cdot (\alpha_2 - \angle BCx) \\ \beta_3 = s \cdot (-s \cdot \alpha_3 - \theta_{ep}) \\ \beta_4 = \alpha_3 + \frac{\pi}{2} \end{cases}. \tag{16}$$

In order to ensure that the landing point will not deviate too far from the target point, and also consider the problems of energy consumption and flight safety, set $R_{ep} \in [R_1, R_2]$ and $\theta_{ep} \in [-\pi, \pi]$.

## 3. Optimization Methods

### 3.1. Gauss Pseudo-Spectral Method

Due to the optimized result of the Gauss pseudo-spectral method, the global optimal-control quantity of the system can be calculated. In this section, the principle of the Gauss pseudo-spectral method is summarized below.

In the Gauss pseudo-spectral method, the state and control variables are discretized at a series of Legendre–Gauss points (LG points), and the LG points are distributed in $[-1, 1]$. Therefore, the real time $[t_0, t_e]$ needs to be transformed to $[-1, 1]$. The transformation form is as follows:

$$\tau = \frac{2t}{t_e - t_0} - \frac{t_e + t_0}{t_e - t_0}. \tag{17}$$

Then, configure $N$ points (LG points) on the open interval $(-1, 1)$, which is defined as $[\tau_0, \tau_1, \cdots, \tau_k, \cdots, \tau_N](k = 1, 2, \cdots, N)$. The points are the zero point of the $N$-order Legendre polynomial. The $N$-order Legendre polynomial can be expressed as:

$$H_N(\tau) = \frac{1}{2^N N!} \cdot \frac{d^N}{d\tau^k}\left[\left(\tau^2 - 1\right)^N\right], \quad N = 0, 1, \cdots. \tag{18}$$

In this paper, considering that the proposed method also involves initial and final value constraints, there are two more points. Define $\tau_0$ and $\tau_e = \tau_{N+1} = 1$ as the initial and final times, respectively. We then have $[\tau_0, \tau_1, \cdots, \tau_l, \cdots, \tau_{N+1}]$. Finally, there are $N + 2$ points.

After discretization, the state values at these LG points are regarded as their true state values. Then, $x(\tau)$ can be obtained by Lagrange interpolation polynomial fitting.

$$x(\tau) \approx X(\tau) = \sum_{i=0}^{N} L_i(\tau) X_i. \tag{19}$$

$$L_i(\tau) = \prod_{j=0,\, j\neq i}^{N} \frac{\tau - \tau_j}{\tau_i - \tau_j}, \quad i = 0, 1, \cdots, N. \tag{20}$$

Similarly, the control variable is discretized at $N$ points (LG points) in $(-1, 1)$, and then $u(\tau)$ can be obtained by Lagrange interpolation polynomial fitting.

$$u(\tau) \approx U(\tau) = \sum_{m=1}^{N} L_m(\tau) U_m, \tag{21}$$

$$L_m(\tau) = \prod_{j=0,\, j\neq m}^{N} \frac{\tau - \tau_j}{\tau_m - \tau_j}, \quad m = 0, 1, \cdots, N, \tag{22}$$

where $L_i$ and $L_m$ denote Lagrange interpolation functions.

Since $x(\tau)$ is only the fitting corresponding to first $N + 1$ nodes, it lacks the terminal state, while $u(\tau)$ lacks the initial and terminal states. Then, the terminal constraint of the system can be expressed as:

$$X_e = X_0 + \frac{t_e - t_0}{2} \int_{-1}^{1} f(x(\tau), u(\tau), \tau) d\tau. \tag{23}$$

After discretizing, we obtain:

$$X_e = X_0 + \frac{t_e - t_0}{2} \sum_{k=1}^{N} \omega_k f(X_k, U_k, \tau_k, t_0, t_e), \tag{24}$$

where $\omega_k$ denotes the integral weight factor.

Then, the initial and terminal control quantities can be expressed as:

$$u(\tau_0) \approx U(\tau_0) = \sum_{m=0}^{N} L_m(\tau_0) U_m. \tag{25}$$

$$u(\tau_e) \approx U(\tau_e) = \sum_{m=0}^{N+1} L_m(\tau_e) U_m. \tag{26}$$

After the derivative of Equation (19), we obtain:

$$\dot{x}(\tau) = \dot{X}(\tau) = \sum_{i=0}^{N} \dot{L}_i(\tau) X_i = \sum_{i=0}^{N} D_{ki}(\tau_k) X_i, \tag{27}$$

where $D_{ki}$ is a differential matrix derived from Lagrange polynomials at point $\tau_k$.

$$D_{ki} = \dot{L}_i(\tau_k) = \sum_{l=1}^{N} \frac{\sum_{j=0, j \neq i}^{N} (\tau_k - \tau_j)}{\sum_{j=0, j \neq i}^{N} (\tau_i - \tau_j)}, k = 1, 2, \cdots, N; i = 0, 1, \cdots, N. \tag{28}$$

Meanwhile, after the discretization of the equation of motion, it can be expressed as:

$$\dot{x}(\tau) = \frac{t_e - t_0}{2} f(X_k, U_k, \tau_k, t_0, t_e). \tag{29}$$

Using the same method, the real-time path constraint can be discretized at point $\tau_k$:

$$C(x_n, U_n, \tau_n, \tau_0, t_e) \leq 0. \tag{30}$$

Then, the objective function also needs to be dispersed at point $\tau_k$:

$$J = \Phi(X_0, X_e, t_0, t_e) + \frac{t_e - t_0}{2} \sum_{n=1}^{K} \omega_n L(X_n, U_n, \tau_n, t_0, t_e), \tag{31}$$

where $\omega_n$ denotes the integral weight factor.

After the above transformation, a series of nonlinear parameter optimization problems can be solved by a sequential quadratic programming algorithm [24]. The optimized flight trajectory is set as the connecting line between some points in the optimization [25]. First, a path planned by a small number of nodes is quickly obtained through the optimization algorithm, and then more nodes are inserted through the interpolation of the trajectory to obtain a higher-precision-flight trajectory.

The position $(x_e, y_e, z_e)$ of the parafoil system after obstacle avoidance can be judged whether it is in the landing area according to $\sqrt{(x_e - x_f)^2 + (y_e - y_f)^2} \leq R_{th}$. Since the parafoil system may still be in the obstacle area after obstacle avoidance, if the parafoil system starts the multiphase homing strategy at this time, it may collide with the obstacles. Therefore, the segmented homing strategy is designed to be started only when the parafoil is in the landing area. Additionally, the Gauss pseudo-spectral method can determine the endpoint of the parafoil system in the obstacle area and produce it in the landing area.

### 3.2. Improved Marine Predator Algorithm

When the parafoil system enters the landing area, according to the relationship between the horizontal distance $l$ from the starting point of the landing area to the target point, the remaining height $h$ after obstacle avoidance and the glide ratio $f$, $l < h \times f$ is divided into feasible areas and $l > h \times f$ is divided into unfeasible areas. For the unpowered parafoil system in this paper, Formula (32) needs to be met in order to accurately reach the destination:

$$h \geq \frac{R_{\text{th}}}{f}. \tag{32}$$

Using the geometric relationship of Formulas (11)–(16), the multiphase homing trajectory planning problem of the parafoil system in the landing area can be transformed into a parameter optimization problem. However, the objective function $J$ of the landing area cannot be explicitly expressed, and there are few parameters to be optimized so that the improved marine predator algorithm can be used for optimization.

The marine predator algorithm (MPA) is a new meta heuristic algorithm to simulate marine hunting behavior proposed by Faramarzi et al. in 2020 [26]. The algorithm is inspired by the movement mode of marine predator and prey. The optimization process is divided into three stages, in which the predators and prey update their positions according to the Levy motion or Brownian motion [27]. At the same time, prey also acts as predator identity, which makes the algorithm more dynamic. Additionally, its unique marine memory storage stage and eddy formation and Fish Aggregating Devices (FADs) effect can further improve the quality of the updated population. In comparison to particle swarm optimization, differential evolution, and other classical algorithms, it has a faster convergence speed and convergence accuracy.

Although the basic MPA algorithm has significant advantages in optimization problems, there are still problems whereby the swarm intelligence algorithm can easily fall into local optimization and a slow convergence speed, which still need to be improved to improve its optimization performance. In order to better improve the optimization accuracy and convergence speed of MPA, this paper combined chaotic opposition initialization with a grouping dimension learning strategy, introduced a t-distribution mutation operator, and proposed a multi-strategy improved MPA, which can significantly improve the optimization accuracy and convergence speed at the same time.

#### 3.2.1. Chaotic Opposition Learning Strategy

Chaotic mapping is a nonlinear theory that has nonlinearity, universality, ergodicity, and randomness [28]. It can traverse all states without repetition in a certain range according to its own characteristics. It can help generate new solutions and increase population diversity in intelligent algorithm optimizations [29]. Therefore, it is widely used. The iterative speed of tent map is fast, and the chaotic sequence is evenly distributed between $[0, 1]$. Its expression is as follows:

$$\lambda_{t+1} = \begin{cases} \lambda_t/\alpha, & \lambda_t \in [0, \alpha) \\ (1 - \lambda_t)/(1 - \alpha), & \lambda_t \in (\alpha, 1] \end{cases}, \quad t = 0, 1, 2, \cdots, T, \tag{33}$$

where $\lambda_t$ is the number of chaos generated in the $i$th iteration, and $T$ is the maximum number of iterations; $\alpha$ is a constant between $[0, 1]$.

In the process of initializing the population by the swarm intelligence algorithm [30], some randomly generated individuals are often distributed in the invalid area and edge area of entering and leaving the optimal solution, which reduces the search efficiency of the population. Using the opposition-based learning (OBL) strategy, introducing a random solution and its opposite solution in population initialization can improve the quality of the initial population better than introducing two independent random solutions.

In this paper, by combining tent chaotic mapping with OBL, a new tent opposition-based learning (TOBL) mechanism was proposed. The mathematical model of TOBL is expressed as:

$$\overline{X}_{i,j} = lb_j + ub_j - \lambda_i \otimes X_{i,j}, \quad i = 1, \cdots, n, \tag{34}$$

where $\overline{X}_{i,j}$ is the $j$ dimensional component of the opposite position of the $i$th prey. The lower and upper bounds of the individual positions are $lb$ and $ub$, respectively. The notation $\otimes$ shows entry-wise multiplications.

The TOBL strategy is equivalent to taking the sum of the upper and lower bounds of the objective function as the center, using the uniform change of tent to dynamically compress the distribution range of the original initial population, and simultaneously trying to make the population uniform.

### 3.2.2. Adaptive t-Distribution

T-distribution is also called student's distribution; the shape of its distribution function curve is closely related to its degree of freedom [31]. In the basic MPA, after the prey updated its position, it was necessary to detect and update the position of the top predator and store it in the marine memory. Next, we considered the influence of fads and further updated the position of the prey.

In order to ensure more effective memory storage, an adaptive t-distribution operator was introduced to mutate the position of prey before simulating the influence of FADs. If the mutated position was better, it replaced the original position. Its mathematical model is expressed as:

$$X_i' = X_i + X_i \cdot t(Iter), \tag{35}$$

where $X_i'$ is the position of the $i$th prey after mutation, and $t(Iter)$ is the t-distribution with the current number of iterations as the degree of freedom.

### 3.2.3. Grouping Dimension Learning Strategy

In the iterative process of the algorithm, some dimensions of some prey positions may have already reached the optimal dimension. Due to the influence of individual dimensions, the fitness of these prey positions becomes worse [32]. In order to survive in the ocean, the prey (or predator) with a poor location needs to learn the predation ability from the prey (or predator) with a good location. Based on this idea, a grouping dimension learning strategy was proposed. The prey affected by FADs was divided into two groups according to the order of fitness. The group with good fitness is called the elite group, and the group with poor fitness is called the learning group.

Because the dimensions of the elite group have their advantages and disadvantages, the position dimension of the elite group is taken as the average value, and each prey in the learning group learns from the average dimension of the elite group. In this strategy, each dimension of each prey in the learning group was compared to the average dimension value of the elite group. According to the priority crossing principle of the large absolute difference, the first $H_1$ corresponding dimensions with large absolute differences were crossed one by one. If the fitness of the prey after the crossing was better, the corresponding dimensions were crossed, and vise versa. Its mathematical model is expressed as:

$$\Delta X_{L,i}^k = \left| X_{L,i}^k - X_{\text{javg}}^k \right|, d \tag{36}$$

$$X_{L,i} = \begin{cases} X_{L,i}^{k,cross}, & f\left(X_{L,i}^{k,cross}\right) \text{ is better than } f(X_{L,i}) \\ X_{L,i} & \text{otherwise} \end{cases}, \tag{37}$$

where $X_{L,i}$ is the $i$th prey position of the learning group, $X_{L,i}^{k,cross}$ denotes the $i$th prey position after crossing the $k$th dimension of the average dimension value of the elite group, $\Delta X_{L,i}^k$ is the absolute difference between the $k$th dimension of the $i$th prey of the learning

group and the $k$th dimension of the average dimension of the elite group, and $X_{javg}^k$ denotes the $k$th dimension of the average value of the elite group.

The larger the value of $H_1$, the more dimensions crossed. For individuals in the learning group, they are close to the average value of the elite group to a great extent, reducing the differences between the individuals in the learning group. At this time, although the average fitness of the population decreased, the convergence speed was accelerated. However, reducing individual differences will bring the risk of falling into local optimization. Even if the convergence accuracy is improved, in comparison to the original algorithm, there is still a certain probability of reducing the optimization accuracy of the improved algorithm. The smaller the value of $H_1$, the less the number of crosses, and the individual difference is small, in comparison to before crossing. Although the individual diversity is maintained, the intersection of fewer dimensions dramatically weakens the ability of the modified algorithm to jump out of the local optimum, and there is also the risk of falling into the local optimum. Moreover, there is minimal crossover, and individuals cannot learn well, which reduces the convergence speed. Therefore, the selection of $H_1$ should be considered in a compromise. In this paper, $H_1$ was selected to be equal to half of the number of individual dimensions.

Because the elite group was relatively close to the global best, it was not suitable for the disturbance variation of all dimensions, which leads to the elite wandering near the optimal solution and affects the convergence accuracy. Therefore, make the prey of the elite group learn from each other and learn from the adjacent prey on the premise of retaining their own dominant dimension. The strategy crossover principle is the same as the prey crossover principle of the learning group, except that the crossover object is replaced by the previous prey adjacent to the prey from the average dimension value of the elite group, and each prey in the elite group is set to take the first $H_2$ adjacent corresponding dimensions with a large absolute difference to cross one by one. In the same way as the $H_1$ selection principle, $H_2$ is equal to half of the number of individual dimensions.

*3.3. Algorithm Flow*

1.  Initialization: set the initial point coordinates of the parafoil system $(x_0, y_0, z_0)$, heading angle $\alpha_0$, target-point position $(x_f, y_f, z_f)$, target-point heading $\alpha_f$, and other information;
2.  Determine whether the position of the parafoil system is within the airdrop area. If not, it ends. Otherwise, proceed to step 3;
3.  Determine whether the position of the parafoil system has entered the landing area. If the parafoil system is in the obstacle area, proceed to step 4. Otherwise, proceed to step 9;
4.  The optimization problem information is extracted, the pseudo spectrum is discretized, and the results are dimensionless;
5.  The discrete parafoil multi-constraint nonlinear programming problem is solved under the current interval grid and interpolation order setting;
6.  If the error meets the requirements, execute step 7. Otherwise, proceed to step 4;
7.  Obtain the optimized trajectory of the obstacle area and increase the control quantity to make the parafoil continue to fly for a period of time, so as to carry out the multiphase homing strategy;
8.  Judge whether the parafoil system position enters the landing area. If the parafoil system is in the landing area, execute step 9; otherwise proceed to step 6;
9.  Conduct trajectory planning in the landing area. Take the sample information entering the landing area as the starting point of the multiphase homing trajectory planning in the landing area and call the optimization algorithm to solve the trajectory parameters in the landing area. Now, the trajectory planning of the parafoil system in a complex environment has been completed.

## 4. Simulation and Analysis

In order to verify the effectiveness of the homing strategy used in this paper, firstly, the multiphase homing strategy planned by the improved marine predator algorithm was compared to other algorithms. The gust interference was considered, and the mountain obstacle constraint was introduced to simulate the proposed combined trajectory planning algorithm, and then the results were compared to the homing strategy of the direct Gauss pseudo-spectrum method.

The states of the parafoil system were set as follows: initial launch point $(x_e, y_e, z_e) = (500, 650, 600)$ (m), target point $(x_f, y_f, z_f) = (0, 0, 0)$ (m), minimum allowable turning radius $R_{min} = 100$ (m), gliding ratio $f = 3$, horizontal speed $v_s = 13.8$ (m/s), vertical velocity $v_z = 4.6$ (m/s), initial angle $\alpha_e = -\pi/4$, and the wind direction near the target point was set in the positive direction along the $x$ axis. Its end angle needed to meet the conditions of upwind alignment, thus $\alpha_f = \pi$ or $\alpha_f = -\pi$. According to these initial conditions, the improved marine predator algorithm, particle swarm optimization algorithm based on logistics chaotic mapping (LPSO), whale optimization algorithm based on chaotic mapping (CWOA), and gray wolf optimizer (GWO) were used as optimization tools to optimize the homing trajectory of the parafoil system in the landing area.

It can be observed from the results in Figure 3 and Table 1 that MSIMPA has a faster iteration speed than the other algorithms, and its final convergence value is second only to the convergence value of LPSO, but the radius optimized by LPSO has a negative value; that is, LPSO may have the result of optimization failure. After comparing the values optimized by several algorithms for the planning of the parafoil system, MSIMPA has a higher accuracy of trajectory results. In the above algorithms, the optimized parafoil system data can realize the segmented homing of parafoil, except for the fact that there are occasional optimization errors in LPSO, resulting in the failure of the segmented homing strategy. Among them, the data optimized by MSIMPA is substituted into the multiphase homing strategy, and the homing trajectory of the parafoil system is shown in Figure 4.

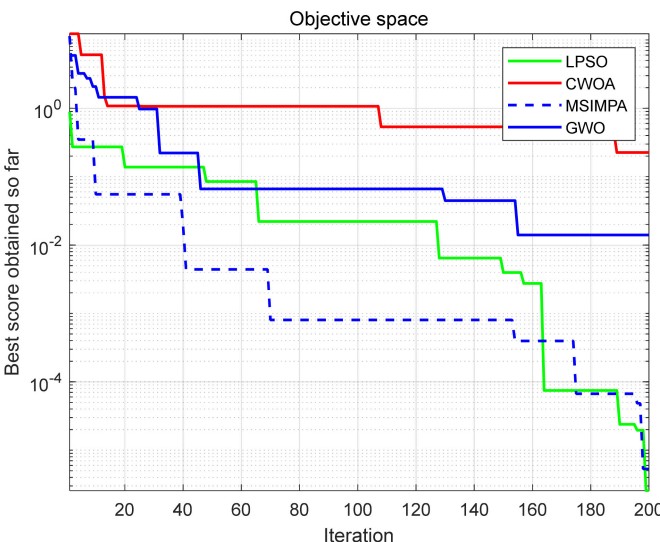

**Figure 3.** The best fitness iteration curve.

**Table 1.** Optimization results.

|  | LPSO | CWOA | MSIMPA | GWO |
|---|---|---|---|---|
| Optimized radius (m) | $-89.313$ | 272.899 | 281.859 | 293.672 |
| Optimization angle (rad) | 1.180 | 0.889 | 0.831 | 0.750 |
| Fitness convergence value | $1.565 \times 10^{-6}$ | 0.226 | $1.491 \times 10^{-5}$ | 0.003 |
| Landing error (m) | - | 0.634 | 0.179 | 0.612 |

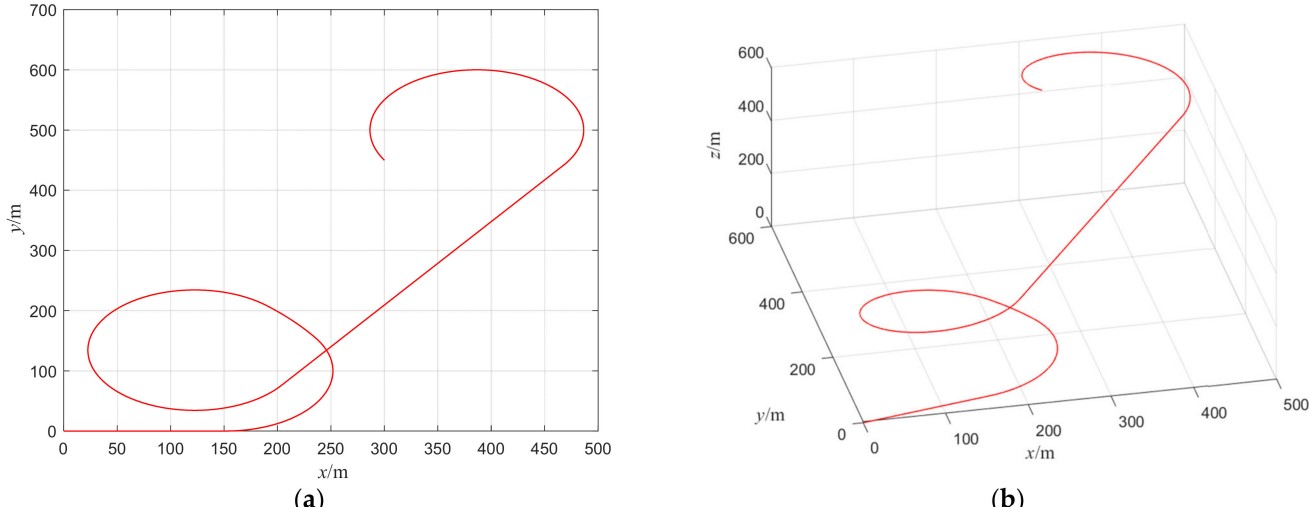

**Figure 4.** Multiphase homing trajectory planning results: (**a**) plane trajectory; (**b**) 3D trajectory.

In order to verify the effectiveness of in the combined homing strategy designed in this paper, the initial position of the parafoil system was set as $(x_0, y_0, z_0) = (400, 3200, 1200)$ m, the target point was set as $(x_f, y_f, z_f) = (0, 0, 0)$ m, the initial angle was set as $\alpha_0 = -\pi/3$, the landing-area radius was set as $R_{th} = 800$ m, the end angle after obstacle avoidance was set as $\alpha_e = -\pi/4$, and the final angle should meet the conditions of upwind alignment, thus $\alpha_f = \pi$ or $\alpha_f = -\pi$. The wind direction near the target point was set to be positive along the $x$ axis; moreover, the location of the center point of the mountain model in this paper is shown in Table 2. At the same time, in order to make the direct homing strategy better compare with the combined homing strategy in this paper, its initial conditions and end conditions were the same as those of the combined homing strategy. Finally, the simulation was conducted according to the above set conditions, and the result of the homing trajectory are shown in Figure 5.

**Table 2.** Mountain model parameters.

| The Ordinal Number | Coordinates (m) | Height of the Mountain (m) | Mountain Radius (m) |
|---|---|---|---|
| 1 | (300, 2300) | 1500 | 400 |
| 2 | (1500, 1500) | 1550 | 400 |
| 3 | (2000, 2500) | 1500 | 400 |
| 4 | (0, 2000) | 1480 | 400 |
| 5 | (500, 1500) | 1500 | 400 |
| 6 | (1000, 3000) | 1490 | 400 |
| 7 | (1500, 0) | 1495 | 400 |

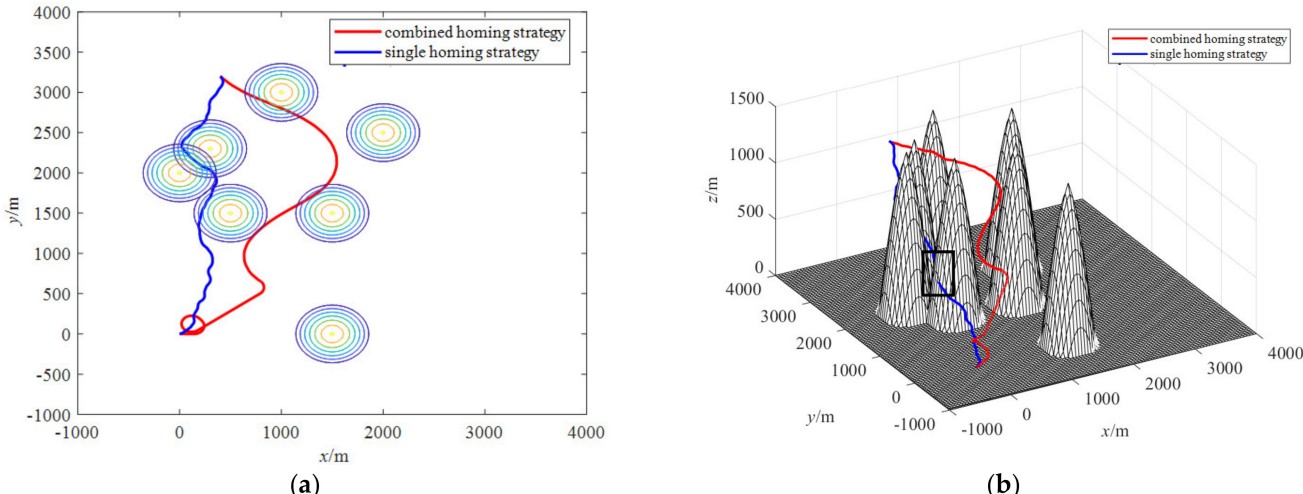

**Figure 5.** Result of trajectory planning of the parafoil system: (**a**) trajectories in the plane; (**b**) 3D trajectories.

From the red line in Figure 5, we can observe that the parafoil is first disturbed by sudden wind, then the heading angle is adjusted to avoid the mountain peak. Finally, it can complete the obstacle avoidance task, reach the landing area, and continues to fly for a period of time, then switch to the subsection homing mode to achieve a windward landing, and its landing point is highly refined. It can be observed that the overall trajectory of the parafoil system is relatively smooth, and the downward trend is relatively gentle. As can be seen from the blue line in the figure, although the parafoil system adopting the direct homing strategy finally reached the target position, it did not completely avoid the obstacles. It can be seen from the black rectangle that the parafoil system passed by the mountain near the landing area.

The control quantities of the two strategies were compared, as shown in Figure 6. It can be seen that the control quantity transformation of the combined homing strategy is relatively gentle, and the control quantity curves are relatively smooth as a whole. Moreover, the parafoil system passes through the obstacle area in more than 80 s, then the control quantity is slowly increased to about 0.138, and then the segmented homing control is conducted. In contrast, the single Gauss pseudo-spectral homing strategy makes the parafoil system stay in the obstacle area for a long time and be more affected by the sudden wind, so there are more oscillations in its control quantity, which means that the parafoil frequently adjusts the control quantity and heading angle in the homing process.

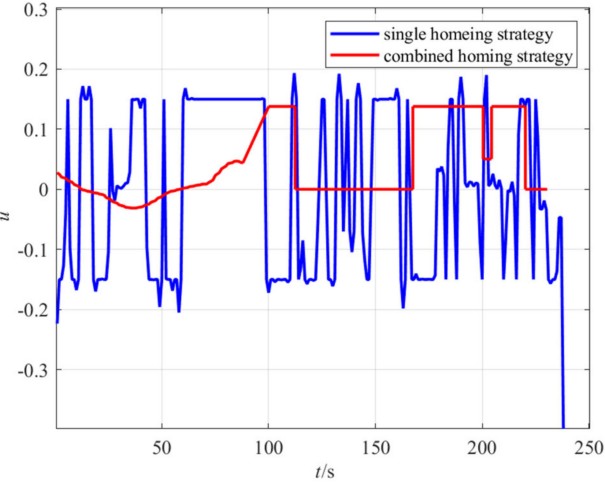

**Figure 6.** Comparison results of control quantities of the parafoil system.

The comparison results of the two trajectory plans are shown in Table 3. It can be seen that the trajectory planned by the single Gauss pseudo-spectral homing strategy under the initial conditions of this paper consumes much energy and may not fully realize the obstacle avoidance function. When approaching the landing area, it may collide with the mountain peak. From the actual project point of views, the control is very difficult, and the whole homing process is challenging to realize. The combined homing strategy is adopted in this paper, because the parafoil enters the landing area after successfully avoiding obstacles, there are few obstacles, and it is close to the target point. At this time, the parafoil system's accurate and gentle landing should be taken as the primary consideration. Therefore, the multiphase homing strategy selected at this time has a segmented constant value, and the control operation is much simpler than the optimal homing, so the control management is relatively small and the simulation time is relatively short. The whole homing process is also easy to realize in practical engineering.

**Table 3.** Comparison of trajectory planning algorithm index results.

|  | Landing-Position Error (m) | Control Management | Simulation Time (s) |
| --- | --- | --- | --- |
| Combined homing strategy | 0.492 | 10.198 | 9.861 |
| Single Gauss pseudo-spectrum homing strategy | 0 | 34.346 | 113.639 |

### 5. Conclusions

In this paper, a combined homing trajectory optimization method was proposed to solve the homing trajectory planning problem of the parafoil system with multiple constraints in a complex environment. To be specific, we divided the parafoil airdrop area into obstacle and landing areas based on the actual airdrop situation. In the obstacle area, the Gauss pseudo-spectrum method with better obstacle avoidance and homing ability was used to realize the obstacle avoidance task of the parafoil system under the influence of sudden wind. For the landing area, a multiphase homing strategy was applied, and the improved marine predator algorithm with better convergence and more stable optimization results was used to optimize the multiphase homing trajectory. Finally, various cases of homing trajectory planning of the parafoil system in complex environments were simulated. The results show that the homing strategy in this paper can meet the requirements of obstacle avoidance and high accuracy of upwind landing under gusts, and the overall trajectory is relatively smooth. The control management of the combined homing strategy is almost one-third of that of the single Gauss pseudo-spectral homing strategy. Moreover, the simulation time is almost one-eleventh of that of the single Gauss pseudo-spectral homing strategy, which highlights the advantages of the combined homing strategy.

However, the studies in this paper did not consider the gust transformation near the landing point, which may affect the trajectory planning in the landing area and cause difficulties in the trajectory tracking control of the parafoil system. In addition, the mountain and parafoil system models used in this paper were relatively simple, but the actual parachute and mountain obstacles should be more complex. For future work, on the one hand, gust interference will be considered in the landing area. On the other hand, more complex parafoil models will be applied for trajectory planning under complex mountain obstacles.

**Author Contributions:** Conceptualization, W.H. and J.T.; methodology, W.H.; software, W.H.; validation, J.W., J.T. and Q.S.; formal analysis, W.H.; investigation, W.H.; resources, J.T. and J.W.; data curation, W.H.; writing—original draft preparation, W.H.; writing—review and editing, J.W., J.T. and Q.S.; visualization, J.W., J.T. and Q.S.; supervision, J.W., J.T. and Q.S.; funding acquisition, J.W., J.T. and Q.S. All authors have read and agreed to the published version of the manuscript.

**Funding:** This research was funded by the National Natural Science Foundation of China, Grant/Award Number: 61963006 and 62003175, the National Science Foundation of Guangxi Province of China, Grant/Award Number: 2018GXNSFAA050029 and 2018GXNSFAA294085.

**Institutional Review Board Statement:** Not applicable.

**Informed Consent Statement:** Not applicable.

**Data Availability Statement:** Not applicable.

**Conflicts of Interest:** The authors declare no conflict of interest. The funders had no role in the design of the study; in the collection, analyses, or interpretation of data; in the writing of the manuscript, or in the decision to publish the results.

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
