# Peer review of "A Combined Homing Trajectory Optimization Method of the Parafoil System Considering Intricate Constraints"

_2673-4052, doi:10.3390/automation3020014_

Round 1

Reviewer 1 Report

The authors consider an extremely important problem of determining the optimal trajectory of parafoil systems among several air obstacles. The influence of complex interference such as wind field and terrain environment is taken into account. A combined trajectory planning strategy of a parafoil system that divides the parafoil airdrop area into an obstacle area and a landing area was proposed. This type of model enables considering the terrain environment surface and several obstacles. Finally, the trajectory of the landing area is designed by means of multiphase homing, and the target parameters are solved by the improved marine predator algorithm that provides a higher accuracy.
There are a lot of shortcomings in a good job that must be corrected.
1. Figure 6 is missing.
2. When defining the variables in the model, metric measures are not given (lines 114-117), eg meters, radians, etc. Some variables are not discussed (t in relation (2)). Actually, it is not known what is being optimized. What are the decision variables?
3. Typos:
a) line 27: "controlled [1]" => "controlled [1]" and many other similar citations;
b) line 87: "2.1. parafoil system Particle Model" => "2.1. Parafoil system Particle Model"
c) line 251: N should be in italics.
The manuscript should be carefully revised, but I should review it again mainly due to the lack of Figure 6.

Reviewer 2 Report

The authors provided interesting research. Nevertheless, I noticed some issues:

Some minor formatting errors like spaces between text and references.

The main aim of the article should be clarified and concretized. In my opinion, now it looks too extended.

It would be nice if the authors could provide a big picture representing the main problem of the research.

The description of the proposed method is a little bit hard to follow. It would be really useful if authors could represent their algorithm in graphical form.

Also, I missed the comparison of the results with similar or at least related research based on similar algorithms. Moreover, provided conclusions are not supported by quantitative results. It would be nice if the authors could provide a discussion on achievements, limitations, and further research directions.

Round 2

Reviewer 1 Report

This version of the manuscript is eligible for publication.

Author Response

Thank you very much for your acknowledgment of our work. 

Reviewer 2 Report

Authors taken into account majority of  provided remarks. In general, I am happy with improvements. Nevertheless, conclusion should be improved by adding some essential numeric values.

Author Response

Thank you very much for your acknowledgment of our work and valuable suggestion. We totally agreed with your comments. In the revised manuscript, we added certain numerical analysis in the conclusion. We hope the revised version could meet your approval.